# Optimizing young tennis players' development: Exploring the impact of emerging technologies on training effectiveness and technical skills acquisition

**Sheng Liu, Chenxi Wu, Shurong Xiao, Yaxi Liu, Yingdong Song** [ID] *

Department of Physical Education and Military Education, Jingdezhen Ceramic University, Xianghu Town, Jingdezhen City, Jiangxi Province, China

* syingdongssx@outlook.com

**Data Availability Statement:** The associated data has been uploaded on the Plos one system and authors has given consent of Publishing the data.

**Funding:** The author(s) received no specific funding for this work.

## Abstract

The research analyzed the effect of weekly training plans, physical training frequency, AI-powered coaching systems, virtual reality (VR) training environments, wearable sensors on developing technical tennis skills, with and personalized learning as a mediator. It adopted a quantitative survey method, using primary data from 374 young tennis players. The model fitness was evaluated using confirmatory factor analysis (CFA), while the hypotheses were evaluated using structural equation modeling (SEM). The model fitness was confirmed through CFA, demonstrating high fit indices: CFI = 0.924, TLI = 0.913, IFI = 0.924, RMSEA = 0.057, and SRMR = 0.041, indicating a robust model fit. Hypotheses testing revealed that physical training frequency ($\beta$ = 0.198, p = 0.000), AI-powered coaching systems ($\beta$ = 0.349, p = 0.000), virtual reality training environments ($\beta$ = 0.476, p = 0.000), and wearable sensors ($\beta$ = 0.171, p = 0.000) significantly influenced technical skills acquisition. In contrast, the weekly training plan ($\beta$ = 0.024, p = 0.834) and personalized learning ($\beta$ = -0.045, p = 0.81) did not have a significant effect. Mediation analysis revealed that personalized learning was not a significant mediator between training methods/technologies and acquiring technical abilities. The results revealed that physical training frequency, AI-powered coaching systems, virtual reality training environments, and wearable sensors significantly influenced technical skills acquisition. However, personalized learning did not have a significant mediation effect. The study recommended that young tennis players' organizations and stakeholders consider investing in emerging technologies and training methods. Effective training should be given to coaches on effectively integrating emerging technologies into coaching regimens and practices.

## Introduction

Training is critical in developing young players by allowing their bodies to build strength and endurance. Routine training helps players improve their skill level, technique, and physical

**Competing interests:** The authors have declared that no competing interests exist.

fitness. According to Kolman [1], sports activities, including tennis, require skills to ensure high performance. Skills in sports involve the ability to perform at high standards effectively by tennis players. Thus, attaining sporting skills requires learning the sport's abilities through training and practice. Skills required for high performance in tennis are categorized into psychological, physiological, and technical domains, each playing a vital role in a player's success; psychological skills include mental toughness, focus, and strategic thinking, enabling players to handle pressure, maintain concentration, and make intelligent decisions during matches [2]. Physiological skills involve physical conditioning elements like strength, endurance, flexibility, and agility, which are crucial for sustaining high-intensity performance, executing powerful shots, and preventing injuries [1]. Technical skills encompass stroke mechanics (service, return, slice, overhead, backhand and forehand strokes), footwork, and tactical play, which are essential for the precise and effective execution of various tennis techniques [2, 3]. The relative impact of these skills on performance is interdependent; psychological resilience can enhance physiological performance by reducing fatigue and improving recovery, while technical proficiency can maximize the benefits of physical conditioning. These skills are utilized in unison to create a holistic framework that supports a tennis player's ability to compete at the highest levels.

Physical conditioning is the other sporting attribute critical for tennis players. Many sports activities are physical, and players require proper strength and conditioning to ensure competition. Physical conditioning thus involves the body's adaptation to strength, flexibility, and endurance through training [4]. Physical conditioning comprises various aspects of athletic training, such as strength training, cardiovascular endurance, flexibility, fitness, and coordination [5]. Strength training focuses on building muscle power essential for explosive movements and endurance in long matches [6, 7]; cardiovascular fitness improves stamina, allowing players to maintain high-intensity performance throughout a match [6, 8]; flexibility training enhances the range of motion, reducing the risk of injuries and improving overall movement efficiency [9]; agility drills increase a player's ability to change direction rapidly, a critical skill in tennis where quick lateral movements are frequent [10, 11]; coordination exercises integrate the neuromuscular system, ensuring smooth and precise movements [12, 13].

The enduring advantages of physical training in tennis are significant; enhanced strength, for example, allows players to hit more powerful shots and contributes to their overall endurance, ensuring they can sustain high levels of play over extended periods [6, 14]. Greater flexibility and agility enable players to reach challenging shots and recover quickly, maintaining their competitive edge; improved coordination results in better stroke mechanics and overall gameplay efficiency [15, 16]. These elements of physical conditioning work together to elevate a player's physical capabilities and contribute to greater resilience against injuries, ensuring they can consistently compete at their highest level for extended periods. Lloyd et al. [17] assert that physical conditioning can be attained through particular training exercises that help players build and develop overall body physical fitness. Similarly, training among young tennis players plays a critical role in building technique. While there is no perfect technique in many sports activities, early preparations through training play an essential role in helping players improve their skills in the game [18].

Training through exercise and workouts among young players helps in developing speed, agility, strength, and coordination. Speed training includes sprint drills, interval training, and plyometric exercises that enhance the player's ability to move quickly across the court, with these exercises improving the reaction time and the explosive power needed for quick sprints and rapid directional changes [19–21]. Agility drills, such as ladder drills, cone drills, and shuttle runs, focus on enhancing a player's ability to change direction swiftly and efficiently, which is crucial for responding to fast-paced rallies and reaching challenging shots [21, 22]. Strength

training integrates weightlifting, resistance training, and bodyweight exercises designed to build muscle mass and increase power, which is essential for generating powerful serves and groundstrokes [23, 24]. Coordination workouts often include balance exercises, hand-eye coordination drills, and functional movement patterns that integrate various muscle groups to work seamlessly; such exercises ensure smooth and efficient movements, improving overall gameplay mechanics [25]. These workouts synergistically strengthen the physical capabilities of young tennis players and improve their on-court performance by enabling them to execute techniques with increased precision and efficacy.

## Challenges of traditional training methods

Traditional tennis sport training methods have long played a critical role in athlete development. These include a variety of foundational practices aimed at developing core skills, physical fitness, and mental acuity; they include on-court drills focused on honing fundamental techniques such as serves, volleys, groundstrokes, and footwork [26, 27]. Traditional routines like regular practice sessions emphasize repetition and precision to build muscle memory and improve consistency; mental conditioning is also integral, involving strategies like visualization, goal-setting, and stress management to boost focus and resilience; match-play simulations and competitive scenarios were also essential to develop tactical understanding and strategic thinking [28, 29]. Coaches provide personalized feedback and continuous assessment to track progress and make necessary adjustments, ensuring a comprehensive approach to athlete development in tennis [30, 31].

However, emerging issues in sports training activities depict challenges inherent in traditional training methods. For instance, traditional sports training methods face the challenges of replicating real-world scenarios [32]. In real-game scenarios, players often face game pressure and fan noise, like the collapse of Martina Hingis at the French Open final against Steffi Graf in 1999; she was a set and two games up when she contested the umpire decision that went against her. This set her against the Parisian crowd, who jeered and booed her all through the match; she eventually lost, and never won another grand slam title after this debacle [33]. Hingis was not mentally prepared to deal with the pressures of the largely vociferous Parisian crowd and the need to maintain her calm not to get on the wrong side of the crowd, which Endo et al. [34] identified can be a deciding factor in games. Traditional training methods need help replicating the real game scenarios, which can lead to differences in players' performance and prevent players from choking on the big stage [35].

The other challenge traditional tennis training activities face is the risk of injuries [36]. Often, traditional training activities include workouts and practice sessions with high physical demands. Engaging players in high physical activities during training increases the risk of injury and potentially leads to players' sidelining for extended periods. Injuries in tennis players can be caused by pressure profiles related to their foot posture and balance abilities. These injuries occur due to specific foot-loading patterns that develop due to repetitive movements during intense tennis training; tennis players are also prone to shoulder injuries such as rotator cuff tears, impact, glenohumeral instability, and tendinopathy in the wrist due to the high amount of spin and speed they generate in their strokes [37–39]. Also, traditional training practices have limited rehabilitation integration and have been identified by studies as affecting recovery times for tennis players [40, 41]. Injured tennis players often require customized sporting practice elements during recovery [42]. Traditional sports training practices often have simple procedure training that does not include customized dynamic training procedures. Limited player rehabilitation integration can lead to slowed recovery from injury and return to top performance.

## Emergence and potential of new technologies in tennis

The sport's continued growth worldwide has led to the need to advance tennis players through training. Various emerging technologies have led to integrating modern training techniques to improve the players' fitness, skill, and agility. Some essential technologies include an AI-powered coaching system, which utilizes artificial intelligence to help coaches analyze and track players' performance [43]. Through AI-based technologies, coaches can gain insights into the players' strengths and weaknesses and consequently make necessary changes in training through personalized training. Virtual Reality (VR) training environment is the other emerging technology in sports training that offers a dynamic and distraction-free environment, allowing players to engage in unlimited practice scenarios [44–46].

Rana andMittal [47] recognized wearable sensors as another promising technology in sports training to enhance athlete performance. Wearable sensors, including functional motions and bio-vital parameters, are often used to monitor workload and health during training [48, 49]. The various sports training technologies present different potential benefits to the player based on their performance. For instance, Wearable Sensor Technology in training offers players improved performance due to the ability to track health and fitness data, which is then used to enhance the individuals' athletic performance [50, 51]. Also, through the technology's predictive patterns, wearable sensor technologies can prevent injuries by predicting risky movements [52, 53]. On the other hand, the use of AI-Powered Coaching Systems in sports training enhances athlete training effectively and skill acquisition by analyzing an athlete's biomechanics and performance statistics to develop customized training [54]. The use of virtual reality (VR) in training also comes with the benefits of skill refinement through the endless practice of similar skills, which helps the players perfect the skills in actual games [55, 56]. Martin [57] disclosed that multi-grand slam winners Novak Djokovic and Andy Murray have benefited from analytics and wearable devices for their success in sports. Gregory [58] stated that Iga Swiatek, a multi-grand slam women's tennis champion, incorporated a hybrid method, where she employs a sports psychologist who uses traditional and emerging technologies to optimize her performance in tennis. No wonder Iga Swiatek is the most successful tennis female tennis player since Serena Williams left the sport.

Various emerging technologies are often used in tennis training. For instance, tennis sport has incorporated AI-powered analytics as an integral part of training and strategizing. AI firms collect tennis statistics of players during matches, which in turn help in training through detailed match report analysis [59]. Players can determine areas of weakness, which helps them train and improve their effectiveness. Analytic technologies are also used in tennis athlete training through SmartCourts that are installed with sensors to capture training data, including ball speed and athlete movements [60]. Telemetry sensors are the other technologies used in tennis training activities. Telemetry sensors involve devices that keep track of the players' technique through keeping track of real-time data analysis [61–64].

## Materials & methods

While technology continues to transform the tennis sports experience through athlete training and performance, past research has explored tennis's complexity by analyzing the use of technology in training. From AI-powered coaching systems to virtual reality training environments and wearable sensors, training technologies play critical roles in skill refinement, improving athlete efficiency, providing safe and controlled training environments, and injury prevention, among other benefits [44, 62, 65]. The systems contribute to skill refinement by analyzing players' biomechanics and performance statistics, providing personalized feedback on techniques such as serve mechanics and footwork [66]. For example, AI can identify and

correct subtle flaws in a player's swing, leading to more consistent and powerful shots [67]; VR training environments allow players to practice under simulated match conditions, honing skills like reaction time and shot selection in high-pressure scenarios; VR can replicate various game situations, enabling players to practice and perfect their responses without actual matches' physical wear and tear [56]. Wearable sensors track real-time data on players' movements and physiological responses, offering insights into aspects like endurance and agility; these sensors can monitor heart rate, muscle activity, and movement patterns, helping players optimize their physical conditioning and injury prevention [68]. Collectively, these technologies enhance specific tennis skills—such as serve accuracy, strategic decision-making and physical resilience—more effectively than traditional training methods alone. Liu [43] articulates that technology has significantly revolutionized tennis training methodologies, leading to the evolution of the sport from a traditional sport to a modern global phenomenon. With the complexity of tennis, players need to improve in various areas during training to enhance their performance, including player position on the court and ball speed and understanding their strength and skill. Dellaserra [69], supported by more recent studies by Cossich et al. [44] and Seçkin et al. [51], affirms that the use of technology in athlete development and coaching continues to become more efficient with the continued utilization of technology and analytics in tennis training.

### Theoretical review

Various theories can be used to explain the concept of emerging technologies in tennis training activities. For instance, the Diffusion of Innovation (DOI) Theory explains how new ideas spread over time throughout a specific social system.

### Diffusion of Innovation Theory

The diffusion of Innovation Theory explains the phenomenon of emerging technologies in tennis training as an idea that gains traction and spreads to society. The concept of diffusion often involves the social process among the populations toward learning and incorporating innovation and technology [70–73]. As tennis evolves from a traditional practice to a modern phenomenon, there is also increased competition through innovation to improve athlete performance. Kellison [74] observes that the diffusion of emerging technologies in sports training may differ based on the target population's characteristics. For instance, the diffusion of emerging technologies in tennis athlete training activities can be categorized into innovators, which include those willing to try new technologies, and the category of early adopters, which include the individuals who embrace changes in technology early. Other categories include the early and late majority and Laggards adopting emerging technologies. The early and late majority categories form most of the population adopting emerging technologies. In the current tennis sport athlete training, the adoption of emerging technologies can be explained in the early majority, as they include individuals who need to see the evidence of the success of the emerging technologies before their willingness to adopt, and those willing to wait for the technology to become widespread before adopting.

### Empirical literature

While empirical research into the impact of emerging technologies in athlete training is still emerging, various studies opine that those technologies significantly influence tennis athletes' training activities, including skill acquisition [16, 65, 75–77]. Training plans for players are critical in improving performance by ensuring balance in training, structure, and progression [78, 79]. Xiao et al. [80] argue that weekly functional training in tennis players improves fitness

and enhances performance. Although weekly functional training plans may not have advantages over conventional training programs, ensuring weekly training programs could outperform conventional training in improving strength and power in tennis players. García-González [81] asserts that weekly training among tennis players helps improve decision-making and performance. Decision-making in tennis is one of the important technical skills that influence the players' performance. Since execution skills are attained through decision-making, weekly training programs are critical in improving performance [10, 82–84]. As a result, the following hypothesis is proposed:

**H1**: *Weekly training plan has a positive and significant influence on technical skills acquisition.*

Researchers have also highlighted that physical training plays a critical role in improving the physical performance of individuals. Among players, frequent physical training is among the training methods employed to gain technical skills [85–88]. Deng [15] is of the opinion that research into the effects of plyometric training (PT) among players indicates significant positive effects on enhancing technical skills, including throwing velocity. Frequent physical training longer than seven weeks indicates positive results in enhancing the acquisition of technical skills. Similarly, Luo and colleagues [89] observed that core training among players could improve skill performance. The baseline for frequent physical training is to improve core skills, which include vital muscle function to ensure agility in the movement of the athlete's body. These assertions led to proposing the following hypothesis:

**H2**: *Physical training frequency has a positive and significant influence on technical skills acquisition.*

Bačić [90] expresses that the automated assessment of tennis swings during coaching plays a vital role in improving performance and safety. Incorporating computational intelligence is critical in improving coaching diagnostics as it helps evaluate the players' technical skills, including "swings" using objective metrics to improve performance including accuracy of shots and consistency as well as speed. The study results indicated that the generation of an artificial intelligence coaching system offers accurate diagnostic feedback that can be used to align the players in customized coaching programs. Han et al. [91] asserts that using AI sensors in tennis training indicates a significant improvement from the traditional tennis serving technique through improved movements. AI sensor analysis uses video analytics and computer technology to guide tennis coaching on avoiding referee penalty levels [92–94]. The following statement was raised for hypothesis three:

**H3**: *AI powered coaching systems have a positive and significant influence on technical skills acquisition.*

Scholars have noted that virtual reality environment training in tennis allows players to practice in real-game scenario environments [95–97]. Le Noury [98] determined that virtual reality technologies in tennis sports training allowed players to develop cognitive skills and technical skill transfer. The virtual reality environment in training offers a safe environment that helps avoid athlete injury. Using AI analytics in VR simulation environments also assists players in transferring technical skills to injury prevention [44, 99]. Bodemer [54] added that the VR environment offers a secure setting for repeating tennis drills, minimising the physical strain typically associated with conventional training approaches. Echoing similar sentiments, Mousavi [100] opines that using virtual reality in training helps significantly improve athlete performance and movement. The virtual scenarios help players adopt different skill levels that can be used in real-life games to improve athlete performance. The following proposed hypothesis is based on the views established by the aforementioned scholars:

**H4:** *Virtual reality training environments have a positive and significant influence on technical skills acquisition.*

Wearable sensors in tennis training use computer analytics to measure biomechanical and athlete functional motions to improve performance [51, 68, 101, 102]. Perri [103] affirms that machine learning algorithms from wearable sensors during tennis training are critical in improving athletes' technical skills, including tennis strokes and movement actions. By analyzing data from wearable sensors, coaches can make improvements through customized coaching programs to improve specific athletes' technical skills. Giménez-Egido et al. [104] proposed that they assist in making specific training modifications based on measured performance and pinpointing technical shortcomings during training sessions. Havlucu et al. [105] emphasized the need of tracking progress to motivate players by showing their advancements over time. Taghavi [106], in support, declares that wearable sensors in tennis training, such as smartwatches, help gather data on technical skills, including tennis stroke detection, leading to improved performance. By gathering performance metrics, players can understand their abilities in real-time and improve their technical skills to improve performance. The following hypothesis is proposed based on these arguments:

**H5:** *Wearable sensors have a positive and significant influence on technical skills acquisition.*

Personalized learning among athletes plays an integral role in determining sport-specific technical skills [107–110]. Koopmann [107] maintains that personal training can be essential for players to attain technical skills. Through personal learning, individuals can demonstrate enhanced capabilities towards sport-specific technical skills, which can be improved to enhance performance. The essence of personal learning in sports is to enhance a more profound understanding of skills to improve the training outcomes and enhance performance. Personal learning can also be understood through the effects of understanding [111, 112]. Bernacki [113] suggests that incorporating learning tasks with out-of-school activities positively affects outcome rating, and personal learning in tennis sports training would positively impact technical skill acquisition. These postulations led to the proposition of the following hypotheses:

**H6:** *Personalized learning has a positive and significant influence on technical skills acquisition.*

**H7a:** *Personalized learning significantly mediates the effect of weekly training on technical skills acquisition.*

**H7b:** *Personalized learning significantly mediates the effect of physical training frequency on technical skills acquisition.*

**H8a:** *Personalized learning significantly mediates the effect of AI powered coaching systems on technical skills acquisition.*

**H8b:** *Personalized learning significantly mediates the effect of virtual reality training experience on technical skills acquisition.*

**H8c:** *Personalized learning significantly mediates the effect of wearable sensors on technical skills acquisition.*

## Conceptual framework

The study aimed to explore the impact of emerging technologies on training effectiveness and technical skills acquisition. For the emerging technologies, three variables are considered—AI-Powered Coaching Systems, Virtual Reality Training Environments, and Wearable

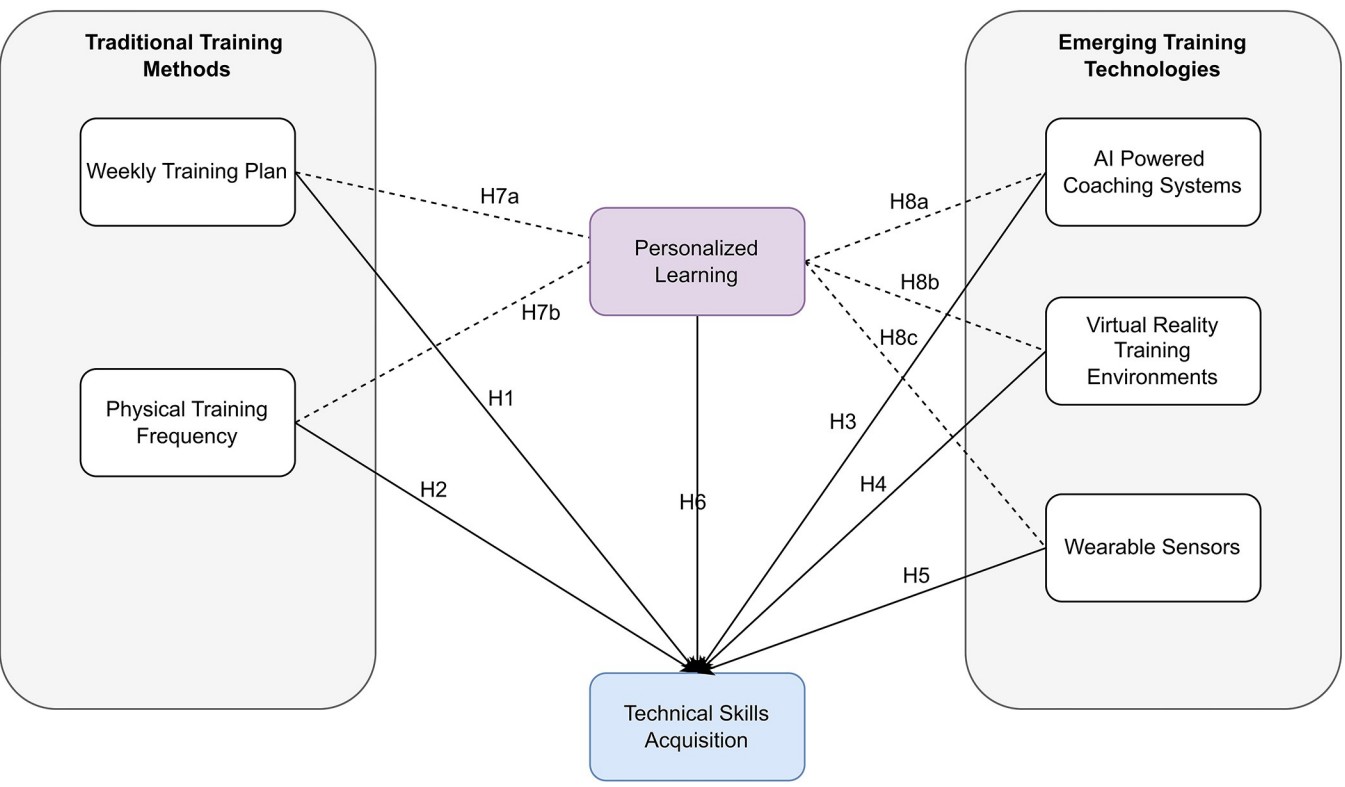

**Fig 1. Conceptual framework.** Source: Compiled by the authors.

Sensors. The variables used for the traditional training methods are weekly training plans and physical training frequency. The mediating variable is personalized learning, while the dependent variable is technical skills acquisition. The information is graphically displayed in Fig 1.

## Methods

The research investigated optimizing young tennis players' development by exploring the effects of emerging technologies and comparing them with traditional training methods. This context fits the quantitative study using a survey research design. The target population from which the participants were obtained were young tennis players under 20. The study focused on the tennis players actively participating in tennis programs in China which number above one million [114, 115]. The Krejcie and Morgan [116] table was used to select the sample size; they recommended using 384 respondents for a study population of 1000000 and above. Thus, a sample size of at least 384 respondents was required to estimate the percentage with no more than 5% error at a 95% confidence level. For uniformity, the sample size was increased to 400. Since the population is large, the study targeted a representative sample of a maximum of 400 young tennis players.

To get the required participants, the researcher used the convenience sampling technique to find the participants who met the required inclusion criteria. They were 20 years old and below tennis players participating in training programs using traditional methods and emerging technologies such as AI-powered coaching systems, virtual reality training, and wearable technology. Convenience was considered most suitable for tennis teams, and coaches would refer to other teams with similar characteristics.

Primary data was collected from the respondents using a structured questionnaire. The questions were structured so the respondents could answer closed-ended questions on their views regarding training aspects, including emerging technologies on training effectiveness and technical skill acquisition. The questionnaire were validated from previous studies; questions on Weekly Training Plan were adapted from García-González [81] and Xiao et al. [80]; questions on Physical Training Frequency were developed from Farrow and Reed [85] and Deng [15]; AI Powered Coaching Systems questions were adapted from Bačić [90] and Bodemer [54]; questions on VR Environments were modified from Bodemer [54], Le Noury [98] and Cossich et al. [44]; Wearable Sensors questions were revised from Seshadri et al. [68], Taghavi et al. [106] and Liu and Zhang [64]; Personalized Learning questions were expounded from Koopmann et al.[107], Larkin and Barkell [108], and Opstoel et al. [109]. The questionnaire comprised two sections–first was respondents' demographic data, and second was questions about latent and observed variables. The questions on latent variables were framed on a Likert scale format, with 1 = strongly disagree and 5 = strongly agree.

The data collection process started by contacting tennis club coaches, leaders, and research participants to introduce the intention of carrying out the study and the objectives the study aims to fulfill. Since the participants were young, informed consent was obtained from all program directors and the participants in this study. An informed consent document was provided explaining the research purpose, procedures, risks and benefits, participant rights, and confidentiality procedures. Verbal consent was obtained from the participants, who were considered minors, in the presence of their coaches and those assigned to be present during the data collection process. We ensured the anonymity of the participants regarding their personal information. To ensure compliance with research ethics, the study received ethical approval from the Institutional Review Board of Jingdezhen Ceramic University, China, with approval number COA No. 0001 in full compliance with the International Guidelines for Human Research Protection such as the Declaration of Helsinki, the Belmont Report, CIOMS Guidelines and the International Conference on Harmonization in Good Clinical Practice (ICH-GCP).

The young tennis players were given printed copies of the questionnaire to complete and return it to the researcher. The data was collected between September 1, 2023, and December 30, 2023. From the target of 400 sample size, 386 responses were successfully filled and returned. Upon cleaning, 374 responses were considered suitable for further analysis.

The data analysis process started with descriptive statistics for demographics, which aimed to understand the study sample's demographic characteristics, such as age, gender, training history, and other relevant demographic variables. The SPSS software was utilized to calculate the mean, standard deviation, frequencies, and percentages. The confirmatory factor analysis (CFA) was used to evaluate whether the observed variables accurately represent the constructs. The model fit using indices such as CFI, TLI, IFIF GFI, and RMSEA was evaluated, as well as the validity and reliability of the study variables. Structural Equation Modeling (SEM) explored the structural relationships between independent, mediating, and dependent variables. The results were presented and interpreted using discussions, tables, and graphs.

## Results

The study findings are reported in this section. The analysis begins with descriptive statistics, followed by model evaluation, and concludes with statistical analysis.

### Demographic data

The first analysis of this research was descriptive statistics of the demographic characteristics of the respondents. The aspects evaluated include age, gender, highest education level, tennis

**Table 1. Demographic information of the respondents.**

| Variables | Categories | Frequency (n) | Percent (%) |
|---|---|---|---|
| Gender | male | 229 | 61.2 |
| | female | 130 | 34.8 |
| | others | 15 | 4 |
| | Total | 374 | 100 |
| Age | Under 10 years | 36 | 9.6 |
| | 10–12 yeas | 47 | 12.6 |
| | 13–15 years | 61 | 16.3 |
| | 16–18 years | 94 | 25.1 |
| | 19–20 years | 136 | 36.4 |
| | Total | 374 | 100 |
| Education Level | Elementary lower | 35 | 9.4 |
| | Elementary upper | 74 | 19.8 |
| | High school | 156 | 41.7 |
| | College | 72 | 19.3 |
| | Undergraduate and above | 37 | 9.9 |
| | Total | 374 | 100 |
| Years playing tennis | less than 1 years | 63 | 16.8 |
| | 1–3 years | 52 | 13.9 |
| | 4–6 years | 109 | 29.1 |
| | 7–9 years | 89 | 23.8 |
| | 10+ years | 61 | 16.3 |
| | Total | 374 | 100 |
| Training days per week | 1–2 days | 105 | 28.1 |
| | 3–4 days | 100 | 26.7 |
| | 5–6 days | 99 | 26.5 |
| | every day | 70 | 18.7 |
| | Total | 374 | 100 |

Source: Compiled by the authors on the basis of the research results.

time, etc. Regarding the respondents' gender, most were male (61.2%), while the females were 34.8%. For the age of the respondents, the majority were 19–20 years (36.4), followed by those aged 16–18 years (25.1%), and then 13–15 years (16.3%). Concerning the education level, the highest proportion was those in high school as highest education level (41.7%), followed by those within upper elementary (19.8%) and then those at college levels (19.3%). The other aspect evaluated was the years the respondents have been playing tennis. The majority of the respondents were those with 4–6 years of playing tennis (29.1%), followed by those with 7–9 years (23.8%) and then those with less than I years (16.8%). The respondents were asked to indicate the training days per week, where the majority of them indicated 1–2 days (28.1%0 followed by 3–4 days (26.7%) and then 5–6 days (26.5%). A summary of the demographic data are presented in Table 1.

## Model fitness evaluation

The model fitness was evaluated by running confirmatory factor analysis (CFA). CFA was used as a statistical technique to test whether the relationships between observed variables and underlying latent constructs are consistent with a hypothesized model. The model quality was assessed by running incremental and absolute fitness indicators. The incremental indices used

were the Comparative Fit Index (CFI), Tucker-Lewis Index (TLI), and Incremental Fit Index (IFI). The required threshold for incremental fitness indices is >0.90 [117]. For the results obtained, the values of CFI = 0.924, TLI = 0.913, and IFI = 0.924. Since they were all above 0.90, the required threshold was satisfied.

The absolute fitness indices used were Root Mean Square Error of Approximation (RMSEA) and Standardized Root Mean Square Residual (SRMR). The required threshold for RMSEA is that < .05 is an excellent fit, while < .08 is a good fit [118]. The statistics for RMSEA = 0.057, while SRMR = 0.041 satisfied the required threshold of a good fit. In addition, the CMIN/DF = 2.208, within the required threshold of <5.0. In addition to fitness indices, the research conducted validity and reliability analysis. The reliability analysis was conducted using Cronbach's alpha and composite reliability, while validity was evaluated using average variance extracted (AVE) and standardized factor loadings. The required threshold for Cronbach's alpha and composite reliability is >0.70 to ensure adequate internal consistency of the scale [119]. The statistics for Cronbach's alpha ranged from 0.78 to 0.88, while the statistics for composite reliability ranged from 0.77 to 0.88, which satisfied the required threshold of reliable internal consistency. The validity of the study was evaluated using standardized factor loadings and average variance extracted (AVE), whose required threshold is >0.50 [120]. AVE assesses the construct validity by examining how much of the variation in the items/questions can be explained by the construct. The values for factor loadings ranged from 0.53 to 0.80, while AVE values ranged from 0.53 to 0.61. This confirmed that the validity of the study constructs was satisfactory, as indicated in Table 2 and Fig 2.

## Hypotheses testing

Structural Equation Modeling (SEM) was adopted to evaluate the study's hypotheses. The specific model adopted was Partial Least Squares Structural Equation Modeling (PLS-SEM). PLS-SEM is a statistical method for analyzing relationships between observed and latent variables. The first analysis of the hypothesis was to evaluate the study's hypotheses expressed as a direct relationship (hypothesis 1 to hypothesis 6). The tests were conducted at a 95% confidence level (0.05 significance level). The results found that the weekly training plan has a positive and insignificant influence on technical skills acquisition (β = 0.024, p = 0.834), rejecting hypothesis 1. The effect size as indicated on Table 3 is very small (0.024), and the high p-value indicates that any observed relationship may be attributed to chance. Physical training frequency was found to positively and significantly influence technical skills acquisition (β = 0.198, p = 0.000); hence, hypothesis 2 was supported. This suggests that physical training frequency has the potential to meaningfully enhance tennis athletes' technical skills. In practical terms, this suggests that increasing the frequency of physical training sessions can substantially enhance players' technical skills. Coaches should consider incorporating more frequent, varied physical exercises to boost overall performance. This could include a mix of strength training, endurance workouts, and flexibility exercises to cover all aspects of physical fitness.

AI-Powered Coaching Systems were found to positively and significantly influence technical skills acquisition (β = 0.349, p = 0.000); accordingly, hypothesis 3 was supported. This point to significant advantage tennis players can derive from involving AI in training environments further boosting their performance. Practically, AI systems can offer personalized feedback, analyze players' performances in real time, and suggest tailored drills and exercises. This can lead to more efficient skill acquisition and refinement. Coaches and training facilities should invest in AI technologies to provide players with advanced, data-driven insights and personalized coaching. Virtual Reality Training Environments were found to positively and significantly influence technical skills acquisition (β = 0.476, p = 0.000); thus, hypothesis 4 was

**Table 2. Model evaluation.**

| Latent variables | Observed Variables | Estimate | AVE | Cronbach's alpha | CR |
|---|---|---:|---:|---:|---:|
| AI | AI1 | 0.71 | 0.53 | 0.85 | 0.85 |
|  | AI2 | 0.71 |  |  |  |
|  | AI3 | 0.76 |  |  |  |
|  | AI4 | 0.74 |  |  |  |
|  | AI5 | 0.73 |  |  |  |
| PL | PL1 | 0.77 | 0.58 | 0.87 | 0.87 |
|  | PL2 | 0.78 |  |  |  |
|  | PL3 | 0.77 |  |  |  |
|  | PL4 | 0.76 |  |  |  |
|  | PL5 | 0.74 |  |  |  |
| PT | PT1 | 0.68 | 0.61 | 0.78 | 0.78 |
|  | PT2 | 0.69 |  |  |  |
|  | PT3 | 0.59 |  |  |  |
|  | PT4 | 0.63 |  |  |  |
|  | PT5 | 0.60 |  |  |  |
| TS | TS1 | 0.53 | 0.60 | 0.78 | 0.77 |
|  | TS2 | 0.58 |  |  |  |
|  | TS3 | 0.64 |  |  |  |
|  | TS4 | 0.68 |  |  |  |
|  | TS5 | 0.73 |  |  |  |
| VR | VR1 | 0.75 | 0.57 | 0.87 | 0.87 |
|  | VR2 | 0.74 |  |  |  |
|  | VR3 | 0.73 |  |  |  |
|  | VR4 | 0.80 |  |  |  |
|  | VR5 | 0.75 |  |  |  |
| WS | WS1 | 0.80 | 0.59 | 0.88 | 0.88 |
|  | WS2 | 0.80 |  |  |  |
|  | WS3 | 0.80 |  |  |  |
|  | WS4 | 0.76 |  |  |  |
|  | WS5 | 0.69 |  |  |  |
| WT | WT1 | 0.77 | 0.57 | 0.87 | 0.87 |
|  | WT2 | 0.80 |  |  |  |
|  | WT3 | 0.74 |  |  |  |
|  | WT4 | 0.75 |  |  |  |
|  | WT5 | 0.71 |  |  |  |

Source: Compiled by the authors on the basis of the research results.

supported, suggesting the highly effective routine of utilizing VR training in skill development. VR can simulate match conditions and scenarios, allowing players to practice and hone their skills in a controlled, immersive environment. This technology can be particularly beneficial for mental conditioning and strategy planning, enabling players to visualize and practice their responses to various in-game situations without actual play's physical wear and tear. Wearable sensors were found to positively and significantly influence technical skills acquisition ($\beta$ = 0.171, p = 0.000); consequently, hypothesis 5 was supported, indicating the utility of integrating wearable devices and technology into tennis players training programs. These devices can track various physiological and biomechanical metrics, providing real-time feedback on

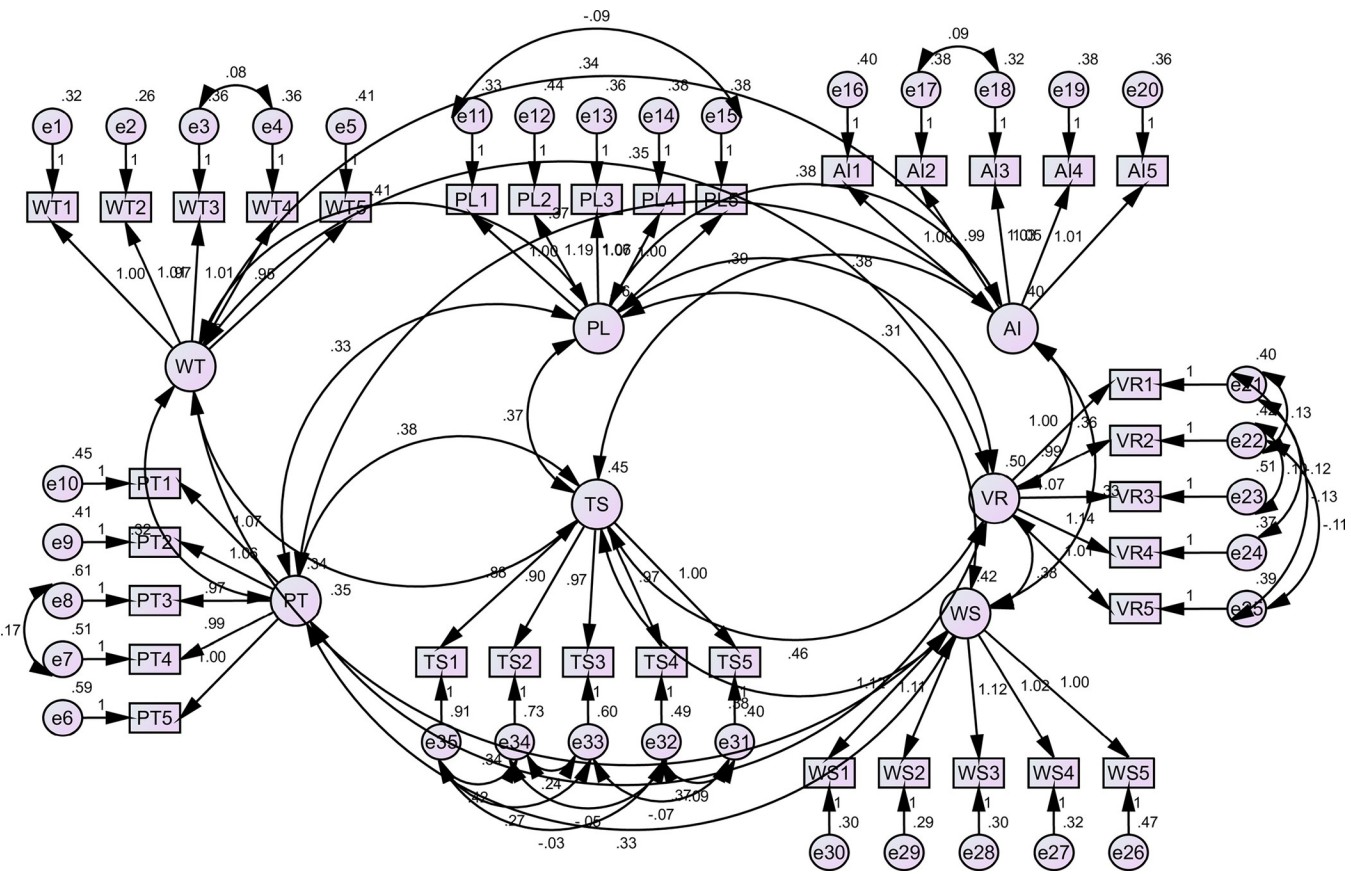

**Fig 2. Validity of the study constructs.** Source: Compiled by the authors on the basis of the research results.

movement efficiency, injury risk, and overall physical condition. Coaches can use this data to adjust training loads, prevent injuries, and optimize performance. For players, wearable sensors can offer insights into their training progress and areas needing improvement. Personalized learning positively and significantly influenced technical skills acquisition (β = -0.045, p = 0.81); therefore, hypothesis 6 was not supported. This may indicate that the method of personalization or the situations in which it was used were ineffective. Table 3 and Fig 3 highlight the hypotheses testing model results.

**Table 3. Hypotheses testing.**

| Hypothesis | Path Relationship | | | Estimate | S.E. | C.R. | P |
|---|---|---|---|---|---|---|---|
| H1 | WT | --> | TS | 0.024 | 0.116 | 0.21 | 0.834 |
| H2 | PT | --> | TS | 0.198 | 0.045 | 4.356 | *** |
| H3 | AI | --> | TS | 0.349 | 0.105 | 3.323 | *** |
| H4 | VR | --> | TS | 0.476 | 0.076 | 6.23 | *** |
| H5 | WS | --> | TS | 0.171 | 0.05 | 3.391 | *** |
| H6 | PL | --> | TS | -0.045 | 0.189 | -0.24 | 0.81 |

AI = AI-Powered Coaching Systems, VR = Virtual Reality Training Environments, WS = Wearable Sensors, WT = Weekly Training Plan, PT = Physical Training Frequency, PL = Personalized Learning, TS = Technical Skills Acquisition

Source: Compiled by the authors on the basis of the research results.

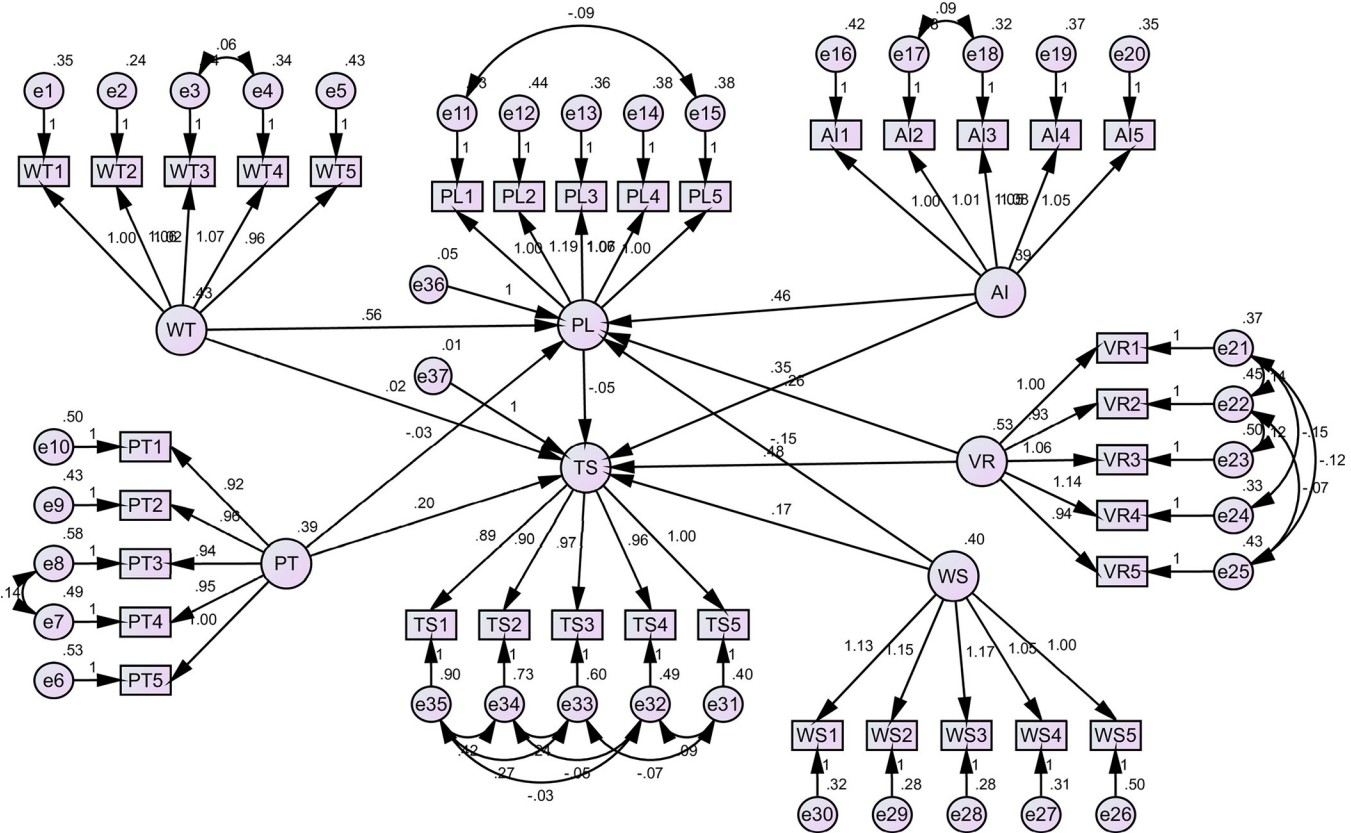

**Fig 3. Hypotheses model results.** Source: Compiled by the authors on the basis of the research results.

In addition to the direct relationship, the indirect relationship was evaluated through the mediating effect of personalized learning. Table 4 assesses mediation hypotheses to determine if Personalized Learning (PL) mediates the relationship between different training techniques and technologies (WT, PT, AI, VR, WS) and Technical Skills Acquisition (TS). This mediation analysis determined if the training techniques impact skills acquisition directly and indirectly through their influence on Personalized Learning. The findings indicated that personalized learning (PL) did not significantly mediate any relationship because all the relationships were

**Table 4. Mediation hypotheses evaluation.**

| Hypothesis | Path | Direct Effects | Indirect Effects | Total Effects | Conclusion |
|---|---|---|---|---|---|
| H7a | WT>PL>TS | 0.024 | -0.025 | -0.001 | Not supported |
| H7b | PT>PL>TS | 0.198 | 0.001 | 0.199** | Not supported |
| H8a | AI>PL>TS | 0.349* | -0.021 | 0.328** | Not supported |
| H8b | VR>PL>TS | 0.476** | -0.012 | 0.464*** | Not supported |
| H8c | WS>PL>TS | 0.171* | 0.007 | 0.178 | Not supported |

Note

*** = significant at 0.01 (99% confidence level)

** = significant at 0.05 (95% confidence level)

* = significant at 0.1 (90% confidence level); AI = AI-Powered Coaching Systems, VR = Virtual Reality Training Environments, WS = Wearable Sensors, WT = weekly training plan, PT = physical training frequency, PL = personalized learning, TS = technical skills acquisition

Source: Compiled by the authors on the basis of the research results.

statistically insignificant. That is WT>PL>TS was insignificant ($\beta$ = 0.007, p>0.05), VR>PL>TS > was insignificant ($\beta$ = -0.012, p>0.05), AI>PL>TS > was insignificant ($\beta$ = -0.021, p>0.05), PT>PL>TS > was insignificant ($\beta$ = 0.001, p>0.05), WT>PL>TS > was insignificant ($\beta$ = -0.025, p>0.05). As a result, hypotheses H7a, H7b, H8a, H8b, and H8c were not supported. The results indicate that PL does not significantly mediate the connection between the analyzed training methods/technologies and Technical Skills Acquisition. The immediate impacts of AI, VR, and PT on skills learning are significant and valuable. However, when factoring in the influence of PL, the overall effect remains mostly unchanged or may even show a minor decline in some instances. While the training approaches are effective, combining them with PL does not improve their performance, especially in this instance.

## Discussion

The survey results revealed that physical training frequency influenced technical skills acquisition significantly compared to traditional training methods. In contrast, the weekly training plan was found to have an insignificant influence. Regarding the frequency of physical training, the results suggest that physical conditioning plays a crucial role in developing technical skills in young tennis players. It highlights the importance of developing physical fitness as a strong foundation for achieving technical sports ability. This affirms Farrow and Reid's [85] findings that physical training is critical in improving general athleticism, which leads to an improved ability to execute technical skills more effectively. Luo et al. [89] in addition notes that including core training into players' routines can enhance skill performance. Frequent physical training aims to enhance core skills, focusing on important muscle function to promote agility in the athlete's body movements. Among the specific types of physical training, high-intensity interval training (HIIT), strength training, agility drills, and endurance exercises were particularly beneficial. HIIT improves cardiovascular fitness and endurance, while strength training builds muscle power essential for powerful strokes and swift movements on the court. Agility drills enhance quick footwork and reflexes, which are crucial for effective court coverage and rapid response to opponents' shots. Regular endurance exercises improve overall stamina, allowing players to maintain peak performance throughout matches. The study suggests that training sessions with moderate to high intensity, conducted at least three to five times per week, correlate more strongly with improved performance. This frequent and varied training regimen ensures comprehensive development of physical attributes, translating into better technical skills and on-court effectiveness.

The study explored the effect of emerging technologies—AI-powered coaching systems, virtual reality training environments, and wearable sensors–on tennis players' technical skills development. The results found that all three emerging technologies influence technical skills development significantly. Virtual reality training environments were found to have the highest influence ($\beta$ = 0.476, p = 0.000) on technical skills development for young tennis players. VR training environments excel due to their immersive and interactive nature, allowing players to simulate real match conditions with high fidelity. This enables focused practice on specific skills in a controlled, distraction-free setting. Unlike AI-powered systems, which primarily analyze and provide feedback, VR offers experiential learning, enhancing muscle memory and spatial awareness. VR training environments replicate real-life training situations, whose high fidelity helps in accelerated skills development through enhanced cognitive and motor learning processes. Compared to wearable sensors, which track and monitor performance metrics, VR actively engages players in scenarios that improve decision-making, reaction times, and situational adaptability. These features collectively make VR training highly effective in skill refinement and technical development.

Scholars contended that incorporating AI sensors in tennis training leads to a notable enhancement in the traditional tennis serving method by refining motions. AI sensor analysis employs video analytics and artificial intelligence to assist tennis coaches in avoiding referee penalties, as discussed in studies by Chmait and Westerbeek [92], Seçkin et al. [51], Li et al [114], and Silva et al. [88]. These findings also agree with Bodemer [54], who indicated that the VR environment provides a safe space for repetitive tennis practices without the physical wear and tear that players experience during traditional training methods. AI-powered coaching systems were a significant influencer in technical skills acquisition. It entails using technologies such as artificial intelligence, computer vision, and neural networks to offer personalized training programs, analyze performance, and real-time feedback. As suggested by Peng and Kim [121], AI-powered systems are helpful in performance analysis by analyzing videos of tennis players in action and identifying technical flaws and areas that may require improvements. Another aspect is that compared to traditional training techniques that depend on post-playing analysis, the AI coaching systems offer immediate feedback, facilitating correcting issues spotted on the spot [93, 94]. This accelerates the learning process and player performance in real-life game scenarios.

Coaches and trainers generally perceive AI-powered coaching systems as valuable tools for enhancing training routines by offering real-time feedback and detailed performance analysis; these systems help in identifying strengths and weaknesses, enabling personalized and data-driven training programs [122, 123]. However, the integration of AI faces some resistance and challenges. One significant concern is the potential over-reliance on technology, which might overshadow the intuitive and experiential knowledge that coaches bring [124]. Additionally, there are challenges related to the initial cost of implementation, the need for technical training to effectively use these systems, and concerns about data privacy and security [123]. Some coaches may also be skeptical about the accuracy and relevance of AI-generated insights compared to traditional coaching methods [125]. To optimize the use of AI in sports training, it is essential to address these concerns by providing proper training for coaches, ensuring cost-effective solutions, and demonstrating the tangible benefits through case studies and success stories.

Wearable sensors were found to have the least effect. Wearable sensors enable real-time tracking during training activities to generate usable data for improvements. Additionally, wearable sensors facilitate movement tracking, analysis, performance monitoring, and optimization. Giménez-Egido et al. [104] suggested that they help in targeted training adjustments through quantified performance and identify technical inefficiencies during training activities. This also aligns with Havlucu et al. [105], who observed that their ability to track progress is vital in motivating players by displaying their improvements over time. Wearable sensors, while crucial for monitoring and feedback, had a lesser impact on technical skills acquisition compared to other technologies due to several factors. One key issue is the potential lag in translating collected data into actionable training insights. Although wearable sensors provide comprehensive data on an athlete's physical condition and performance, the sheer volume of data can be overwhelming and may require significant time and expertise to analyze effectively [125]. Coaches and athletes might not immediately understand or utilize this data optimally, leading to underutilization of the insights these devices offer. Additionally, the data from wearable sensors often need to be integrated with other training information to provide a holistic view of performance, which can be complex and time-consuming. To enhance the impact of wearable sensors, improving data analysis tools and training coaches to interpret and apply the insights quickly and effectively could bridge the gap between data collection and actionable improvements in training.

Several aspects could be pointed out as to why the weekly training plan was insignificant in technical skills acquisition. First, it suggests that developing a training plan without proper

consideration of the quality of training would not bear desirable results as well as the flexibility needed to adapt to athletes' evolving needs and performance level. It emphasizes that the quality of training, including personalized feedback and specific skill-focused drills, is more crucial than the quantity of training sessions. Tennis players, particularly young ones, often require dynamic and responsive training schedules that can be adjusted based on their immediate progress, fatigue levels, and skill acquisition rates. A rigid weekly plan might not accommodate these aspects. The findings of the insignificant effect of personalized learning contradict some literature findings. For instance, Zhang [29] and Koopmann et al. [107] support the effectiveness of personalized learning approaches in sports training. The difference could be traced from the fact that the ability to influence technical skills acquisition depends on how the personalized training is implemented and integrated with other training components. The effects of the effectiveness of personalized learning hinges on how well it is implemented. If the personalized learning strategies were not adequately tailored to the athletes' specific needs and current skill levels, or if the implementation lacked consistency and depth, the expected benefits might not materialize. Personalized learning requires detailed and ongoing assessments to tailor the training accurately, which might have been a challenge in this study. Personalized learning plans can sometimes be challenging for coaches and players. The potential positive effects could be diluted if the athletes or coaches found it challenging to fully engage with the personalized plans due to their complexity or insufficient resources (e.g., time and technological tools). Also, personalized learning needs to be well-integrated with other training components to be effective.

The results show that the mediation effects were not supported, indicating that implementing personalized learning requires adjustments to capture its potential mediating effects better. There may be a need for more robust algorithms and technologies that provide highly personalized training programs to be integrated based on a detailed analysis of players' performance, learning styles, and progress, which includes AI-driven adaptive learning systems that modify training content and difficulty in real-time. Another method of improving the effects of mediation is integrating personalized learning with other AI, VR, and wearable sensors for better individual tracking, health, and performance data. Training sessions should be customized in content and structure, allowing flexibility in timing, intensity, and type of exercises based on the player's evolving needs and responses. Future studies can mitigate this by using a mixed-method approach incorporating qualitative and quantitative data or using control groups that do not receive personalized learning interventions to isolate and measure the effects of personalization more clearly and also by including control groups that do not receive personalized learning interventions to isolate and measure the effects of personalization more clearly.

## Conclusions

The study compared the emerging technologies in tennis training programs with traditional training programs, examining how the new technologies enhance training efficiency and the acquisition of technical skills. The research relied on primary data from young (20 years and lower) tennis players. The traditional training effects were assessed by weekly training plan (hypothesis 1) and physical training frequency (hypothesis 2). The results indicated that physical training frequency significantly influences technical skills acquisition in young tennis players. With regard to emerging technologies, the research evaluated the effect of AI = AI-powered coaching systems (hypothesis 3), virtual reality training environments (hypothesis 4), and wearable sensors (hypothesis 5) on technical skills acquisitions. The results revealed that the three emerging technologies significantly and positively affect young tennis players' technical skills acquisition. The research indicated that emerging technologies are vital in using

artificial intelligence, computer vision, and neural networks to offer personalized training programs, analyze performance, and provide real-time feedback. For instance, wearable sensors help target training adjustments through quantified performance and identify technical inefficiencies during training activities. The VR environment also provides a safe space for repetitive tennis practices without the physical wear and tear that players experience during traditional training methods. The effect of personalized learning on technical skills acquisition was found to be insignificant. The mediating effect of personalized learning on both traditional methods and emerging technologies was found to be insignificant.

Several recommendations were developed from theoretical and managerial perspectives from this research. They serve as future directions to maximize the benefits of emerging technologies while addressing potential challenges to enhance training effectiveness and technical skill acquisition among young tennis players. First, this research advocates that young tennis player organizations and stakeholders should consider investing in emerging technologies and training methods such as AI-powered coaching systems, wearable sensors, and VR training environments. These technologies have great potential for improving training effectiveness and technical skills acquisition. Secondly, coaches should be given effective training on effectively integrating emerging technology into their coaching practices. For instance, how to use AI systems, read and interpret data from these technologies and apply it for improvement. Coaches should develop a habit of applying the insights for personalized training young tennis players. For young tennis players, this study recommends that they be open to using new technologies to improve their skills and overall performance. More importantly, they incorporate the feedback and insights from these technologies during their training sessions.

There were some limitations observed in the course of this research, one is the gender imbalance of the study population with 61.2% male respondents. This could affect the generalization of the study findings. The skew could impact the interpretations of the effectiveness of the different training technologies in different instances; boys and girls may respond to training methods differently on the account of psychological and biomechanical differences, hormonal differences can inhibit muscle recovery and adaptations potentially leading to different results in terms of physical conditioning and skill acquisition. Different outcomes may also be encountered by the different genders when interacting with technology in different endeavors based on extensive documentation in prior studies [105, 126–132]. Future studies should take gender representation into account in the study design and data collection technique. Future studies could also consider investigating the effects of integrating personalized learning with other AI, VR, and wearable sensors to ascertain how they contribute to an athlete's individual tracking, health, and performance data, and how they all correlate into improving athlete performance. There is also a very important need to utilize qualitative studies to understand the feedback from participants about their insights on personalized learning, and how it can best be deployed to address individual idiosyncrasies.

## Supporting information

**S1 Data.**
(ZIP)

## Acknowledgments

The authors would like to thank the survey participants and the reviewers for their contributions to the research.

## Author Contributions

**Conceptualization:** Chenxi Wu.

**Data curation:** Chenxi Wu, Yingdong Song.

**Investigation:** Shurong Xiao, Yingdong Song.

**Methodology:** Yingdong Song.

**Resources:** Yingdong Song.

**Software:** Shurong Xiao.

**Validation:** Yaxi Liu.

**Visualization:** Yaxi Liu.

**Writing – original draft:** Sheng Liu.

**Writing – review & editing:** Yaxi Liu, Yingdong Song.

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
