## [Decision Letter · Decision Letter 0]

13 May 2024

PONE-D-24-10897Optimizing Young Tennis Players Development: Exploring the Impact of Emerging Technologies on Training Effectiveness and Technical Skills AcquisitionsPLOS ONE

Dear Dr. Song,

Thank you for submitting your manuscript to PLOS ONE. After careful consideration, we feel that it has merit but does not fully meet PLOS ONE’s publication criteria as it currently stands. Therefore, we invite you to submit a revised version of the manuscript that addresses the points raised during the review process.

We look forward to receiving your revised manuscript.

Kind regards,

Yaodong Gu

Academic Editor

PLOS ONE

Journal Requirements:

3. In the ethics statement in the Methods, you have specified that verbal consent was obtained. Please provide additional details regarding how this consent was documented and witnessed, and state whether this was approved by the IRB

"No authors have competing interests"

Additional Editor Comments:

Please check the question raided by the reviewer in methods part, the subjects shall be reported reliable data.

Reviewers' comments:

Reviewer's Responses to Questions

**Comments to the Author**

1. Is the manuscript technically sound, and do the data support the conclusions?

Reviewer #1: Yes

Reviewer #2: Yes

2. Has the statistical analysis been performed appropriately and rigorously? 

Reviewer #1: Yes

Reviewer #2: Yes

3. Have the authors made all data underlying the findings in their manuscript fully available?

Reviewer #1: Yes

Reviewer #2: Yes

4. Is the manuscript presented in an intelligible fashion and written in standard English?

Reviewer #1: Yes

Reviewer #2: Yes

5. Review Comments to the Author

Reviewer #1: Thank you for the opportunity to review your manuscript titled "Optimizing Young Tennis Players' Development: Exploring the Impact of Emerging Technologies on Training Effectiveness and Technical Skills Acquisition." This study addresses an important area in sports science, focusing on how emerging technologies can be integrated into the training programs of young athletes to enhance their performance.

General Comments:

Your manuscript is well-structured and introduces an innovative angle on sports training technologies. However, the manuscript would benefit significantly from a deeper analysis and clearer exposition of some key aspects:

Statistical Analysis and Results Interpretation:

The results section provides a broad overview of findings but lacks depth in the interpretation of these results. It would be beneficial to include a more detailed discussion on how these findings compare with existing literature, especially concerning the effectiveness of different technologies.

Methodological Rigor:

More detailed information on the methodological design would enhance the credibility of the study. Specifics regarding the participant selection process, the configuration of technologies used, and the statistical methods applied should be clearly articulated.

Ethical Considerations:

Given that your study involves young participants, a more thorough explanation of the ethical considerations and consent procedures would be appropriate. Please clarify how informed consent was obtained from participants or their guardians and discuss any ethical approvals obtained for the study.

Limitations and Future Research:

While some limitations are briefly mentioned, a more comprehensive discussion would provide a better context for the findings. Additionally, suggestions for future research could be more specifically tied to the limitations and findings of the current study.

Research and Publication Ethics:

No concerns about dual publication or research ethics have been identified based on the review. However, ensuring that all data presented is original and not previously published is crucial. If any part of the study overlaps with previous publications, those should be cited appropriately.

Overall Recommendation:

I recommend that the manuscript be considered for publication after major revisions. The suggested changes will strengthen the argument, improve the readability, and enhance the manuscript’s contribution to the field of sports science and athletic training.

These comments are intended to guide you in strengthening your manuscript and do not detract from the importance and timeliness of your research. I look forward to seeing the revised version of your work.

Reviewer #2: Review comment

This manuscript entitled “Optimizing Young Tennis Players' Development: Exploring the Impact of Emerging Technologies on Training Effectiveness and Technical Skills Acquisition” primarily aimed to evaluate the impact of emerging technologies on the effectiveness of training and technical skills acquisition among young tennis players. The results of this study provide guidance for human movement science and sports science. While it is a very interesting topic. But I think this manuscript has some flaws to fill in before it can be published in a journal. There are several questions should be addressed, which list below. I give a major revision for this manuscript.

Specific comments

1. In the Abstract part, " The rapid advancement in sports science and technology has replaced traditional training methodologies..." The statement implies a complete replacement of traditional methods. Can the authors clarify if any traditional training methodologies are still effective or relevant in current sports science, particularly in tennis, despite the advancements mentioned.

2. In the introduction part, “According to Kolman , sports activities, including tennis, require skills to ensure high performance.” The introduction discusses skills required for high performance in tennis. Could the authors expand on how these skills are categorized (e.g., psychological, physiological, technical) and their relative impact on performance.

3. “Training through exercise and workouts among young players helps in developing speed, agility, strength, and coordination.” Please use more then one references to support this sentence.

4. “Telemetry sensors involve devices that keep track of the players' technique through keeping track of real-time data analysis” Symmetry training is also an important ability in tennis, and the reviewers suggest that the authors add relevant references. (DOI: 10.5334/paah.215).

5. In the Materials and methods part, Given the quantitative approach, how do the authors ensure that the subjective nature of self-reported data does not bias the outcomes, particularly in assessing the effectiveness of AI and VR technologies in training.

6. In the discussion part. "Virtual reality training environments were found to have the highest influence..." The manuscript highlights the significant impact of VR environments. Could the authors speculate on the potential long-term effects of VR training on athletes' performance, particularly regarding skill transfer to real-world settings.

7. In the Conclusion part, "The research indicated that emerging technologies are vital..." In concluding that emerging technologies are vital, how do the authors address the challenges or limitations that these technologies present in practical training settings.

6. PLOS authors have the option to publish the peer review history of their article (what does this mean?). If published, this will include your full peer review and any attached files.

Reviewer #1: No

Reviewer #2: **Yes: **Zixiang Gao

---

## [Author Response · Author response to Decision Letter 0]

27 Jun 2024

Response to Editor and Reviewer comments 

EDITOR REVIEW COMMENTS

Authors response 

The authors thank the editor and the reviewers of our manuscript titled "Optimizing Young Tennis Players' Development: Exploring the Impact of Emerging Technologies on Training Effectiveness and Technical Skills Acquisition" for the great review. We will attempt to respond to the questions raised by the editor and the reviewers.

Specific Comments: 

Abstract：Could you expand on the results in the abstract to include specific statistical details? This would provide a clearer and more comprehensive picture of how each technology influenced technical skills acquisition and the extent to which personalized learning did not have a significant mediation effect. 

Response: The abstract has been rewritten to include specific statistical details including the model fitness results and the hypotheses testing.

Introduction：It is recommended to reorganize the introduction. Could expanding the discussion on existing research and literature more clearly demonstrate the specific knowledge gaps your study addresses? There is a need to more tightly connect the shortcomings of traditional methods with the advantages of new technologies. Please clearly indicate how each emerging technology overcomes specific limitations of traditional training, including detailed application examples, thereby enhancing the article's arguments. 

Response: The section on “Challenges of traditional training methods” and “Emergence and potential of new technologies in tennis” discussed examples of the shortcomings of traditional training methods and how emerging technologies is filling the gap covering the limitations of traditional training methods and how they have been enhanced bt new technologies in tennis. Specific examples have been added of tennis players who have benefitted from emerging technologies.

1. "Physical conditioning is the other sporting attribute critical for tennis players. Many sports activities are physical, and players require proper strength and conditioning to ensure competition." What specific elements does 'physical conditioning' comprise? How does it impact the athletic performance of tennis players? 

Response: the elements of physical conditioning have been added in in the introduction, including how it impacts athletes’ performance.

2. "Training through exercise and workouts among young players helps in developing speed, agility, strength, and coordination." Could you specify which types of 'exercise and workouts' are included here?

Response: paragraph has been included to include types of exercises and workouts that help tennis players develop speed, agility, strength and coordination. 

3. "Traditional tennis sport training methods have long played a critical role in athlete development." Could you detail what encompasses traditional training methods? Please include necessary references.

Response: Traditional training methods include fundamental techniques such as serves, volleys, groundstrokes, and footwork in regular practice sessions that emphasize repetition and precision to build muscle memory and consistency. These have been outlined including supporting references (Gomez-Pinilla & HillmanN, 2013; Nugroho et al., 2023; Ericsson & Harwell, 2019; Zhang, 2024; Mason et al., 2020; Otte et al., 2020).

4. "However, emerging issues in sports training activities depict challenges inherent in traditional training methods." Could you provide a detailed explanation of "emerging issues," offering specific examples or data that support these challenges in traditional training methods?

Response: The emеrging issues have been further explained with specific examples relevant to the discussion like the Martina Hingis example and the report by Williams and Wigmore (2020) highlighting challenges athletes’ face during games including the pressure from fans and the game itself. 

5. "Traditional training methods need help in replicating the real game scenarios, which can lead to differences in players' performance." You noted challenges with replicating real game scenarios using traditional training methods. Could you elaborate on how these limitations specifically affect the skill development of young tennis players?

Response: The example used was the Martina Hingis meltdown in the final of the 1999 French open where she lost after challenging the umpire and getting on the wrong side of the crowd. She never won another grand slam in her career. This is an example of where the pressure affected the growth of a 19 year old tennis prodigy.

6. "High physical activities during training increases the risk." Could you describe what types of injury risks are associated with high-intensity physical activities?

Response: We have added examples of the injuries “…foot-loading patterns that develop due to repetitive movements during intense tennis training; tennis players are also prone to shoulder injuries such as rotator cuff tears, impact, glenohumeral instability, and tendinopathy in the wrist due to the high amount of spin and speed they generate in their strokes.”

7. "Also, traditional training practices have limited rehabilitation integration." "Limited player rehabilitation integration can lead to slowed recovery from injury and return to top performance." How does traditional training impact rehabilitation integration? Is there any data or research to support this?

Response: Traditional recovery procedures often involve generalized procedures for anyone with suh an injury compared to customized dynamic procedures that are created specific for the athlete. A lot of data is considered before recommending a recovery strategy that is optimized for the athlete as Ayala et al. (2016) and Yildiz et al. (2019) identified. 

8. "Some essential technologies include an AI-powered coaching system, which utilizes artificial intelligence to help coaches analyze and track players' performance." Could you provide specific examples of how AI systems analyze and track performance, and how this data is subsequently utilized to develop training plans?

Response: “Some essential technologies include an AI-powered coaching system, which utilizes artificial intelligence to help coaches analyze and track players' performance (Liu, 2021). Through AI-based technologies, coaches can gain insights into the players' strengths and weaknesses and consequently make necessary changes in training through personalized training.” This already answers your question. The study by Liu (2021) provides data that show how an AI is used by coaches to analyze player strengths and weaknesses and know where to optimize training to improve performance.

9. "Virtual Reality (VR) training environment is the other emerging technology in sports training that offers a dynamic and distraction-free environment." Can you provide examples of how VR technology creates a "distraction-free" environment and how this specifically benefits technical skill training?

Response: VR environments are simulations, thus, athletes can see themselves in action and assess their own strengths and weaknesses, and also try out new methods to see how it would turn out without risk to themselves in any way. VR environment are “distraction free” because only the athlete or those connected to the system will have access to the data the athlete is examining or watching the athlete perform in the virtual world where they can practice without external limitations.

10. "Wearable Sensor Technology in training offers players improved performance due to the ability to track health and fitness data, which is then used to enhance the individuals' athletic performance." How does Wearable Sensor Technology enhance athletic performance through tracking health and fitness data? Please provide a detailed explanation.

Response: Wearable devices monitor health and fitness data. This includes sleep times, steps taken in a day, sprint race times, heart rate, blood pressure etc. These devices work round the clock monitoring player performance during training and in real-game scenarios, which can then be analyzed to ensure that training performances are replicated during match plays. Any deviation is then investigated to understand the reason. For example, the coach can work with the player to ensure that they maintain the same blood pressure during training and game times, any deviations will lead to studying the data to understand why the blood pressure rises during games for example while lower during training situations. The coaches will then ensure to get the player in the same condition at all times to replicate their performance in training during match plays.

Methods

1. "From AI-powered coaching systems to virtual reality training environments and wearable sensors, training technologies play critical roles in skill refinement..."Could you provide more specific examples of how each technology—AI systems, VR environments, and wearable sensors—concretely contributes to skill refinement in tennis training? Are there particular skills these technologies target more effectively.

Response: Specific examples have been provided showing how AI systems, VR and wearable sensors impact players and the particular skills these technologies improve. References were also added to support them. 

2. "The diffusion of Innovation Theory explains the phenomenon of emerging technologies in tennis training as an idea that gains traction and spreads to society." How does the Diffusion of Innovation Theory specifically apply to the adoption rates of these technologies among tennis coaches and athletes? Can you elaborate on how the characteristics of early adopters versus laggards manifest in this particular sports training context?

Response: The Diffusion of Innovation Theory, applied to adopting emerging technologies in tennis training, illustrates how tennis coaches and athletes progressively embrace AI-powered coaching systems, VR environments, and wearable sensors. Early adopters, typically innovative and risk-taking coaches, and athletes, quickly integrate these technologies to gain a competitive edge, utilizing AI for personalized training plans, VR for immersive practice sessions, and wearable sensors for detailed performance analytics. They value the potential for enhanced training efficiency and improved performance. In contrast, laggards are more traditional, skeptical of new technologies, and slower to change their established training routines, mainly due to the inability to meet the cost of the new technologies. They may rely on proven, conventional methods and adopt new technologies only when they become mainstream and demonstrate clear, widespread benefits. This disparity results in a gradual shift in training practices, with early adopters leading the way and demonstrating the advantages, eventually convincing laggards to follow suit as the technologies prove their efficacy in improving athletic performance. This can also be attributed to the fact that developing nations hardly have any top players in tennis; they are still relying on traditional methods, making it hard for their players to break into the top-tier of tennis.

3. "Various studies opine that those technologies significantly influence tennis athletes' training activities..."Are there contrasting studies that may not support the effectiveness of these technologies in enhancing tennis training? Including a range of studies might provide a balanced view and strengthen the argument for the adoption of these technologies.

Response: There are no studies disputing the impact of AI. It will be like going backwards to be against the use of AI, because in the real sense, AI is not really making the player better by giving them strength overnight, what it does is tell them what they are doing wrong and where they need to improve. For example, if the data for a 5-year period show that every morning a player takes tea without milk, he wins his tennis matches, AI will only point this data out and recommend that the player takes tea without milk on game days or sleeping patterns too. AI in this instance is more about player data and helping the player understand themselves better. Thus, more information will always lead to better outcomes. 

4. "The research investigated optimizing young tennis players' development... The study targeted a representative sample of a maximum of 400 young tennis players." How was the sample size of 400 young tennis players determined to be sufficient for this study? What statistical power analysis was conducted to ensure this sample size could adequately detect the expected effects?

Response: The authors used the Krejcie and Morgan (10970) sample size table that recommends using 384 sample size for populations of one million and above. For uniformity, the authors increased the sample size to 400.

5. "Primary data was collected from the respondents using a structured questionnaire... The questions on latent variables were framed on a Likert scale format." How were the questionnaire items developed and validated to ensure they accurately measure the constructs of interest, such as training effectiveness and technical skill acquisition? Were any pilot tests conducted to refine the questions before the main data collection?

Response: The questionnaire items were developed already developed scales discussed in the empirical literature; the Weekly Training Plan were adapted from García-González (2014) and Xiao et al. (2023); questions on Physical Training Frequency were developed from Farrow and Reed (2010) and Deng (2023); AI Powered Coaching Systems questions were adapted from Bačić (2018) and Bodemer (2023); questions on VR Environments were modified from Bodemer (2023), Le Noury (2021) and Cossich et al. (2023); Wearable Sensors questions were revised from Seshadri et al. (2019), Taghavi (2019) and Liu and Zhang (2022); Personalized Learning questions were expounded from Koopmann et al. (2020), Larkin and Barkell (2022), Opstoel et al. (2019) and Koopmann (2020). Because the scales had been pretested in the previous studies, there was no need to pilot test the questionnaire.

6. "Since the participants were young, informed consent was obtained from all participants in this study... We obtained verbal consent." Given the involvement of minors in the study, could you explain why verbal consent was deemed sufficient? What additional measures were taken to ensure the ethical treatment of these participants, especially concerning their comprehension of the study's aims and their rights?

Response: Verbal consent was required to ensure the participants were comfortable answering the questions. The verbal consent was in addition to the ethics review board approval received from the study, and also explaining to the center managers and coaches the aim of the study. 

7. "The confirmatory factor analysis (CFA) was used to evaluate whether the observed variables accurately represent the constructs... Structural Equation Modeling (SEM) explored the structural relationships." Can you provide details on the model fit indices used in CFA and SEM, and what benchmarks were used to determine a good fit? How the models were adjusted based on the initial outcomes of these analyses?

Response: The details of the model fit indices were provided in “Model Fitness Evaluation,” where the benchmarks for each index were identified and the supporting references for each benchmark included. There was no need for adjustments as the models were all good fits satisfying the threshold of being above 0.90. 

RESULTS

1. The demographic distribution seems heavily skewed towards male participants (61.2%). Could this gender imbalance influence the generalizability of your results? How might this skew impact the interpretations of the effectiveness of various training technologies? 

Response: The gender imbalance, with 61.2% male participants, could influence the generalizability of the results. This skew could affect the interpretations of the effectiveness of various training technologies in several ways:

Training Response Variability: Men and women may respond differently to training stimuli due to physiological and biomechanical differences. For

---

## [Decision Letter · Decision Letter 1]

15 Jul 2024

Optimizing Young Tennis Players Development: Exploring the Impact of Emerging Technologies on Training Effectiveness and Technical Skills Acquisitions

PONE-D-24-10897R1

Dear Dr. Song,

We’re pleased to inform you that your manuscript has been judged scientifically suitable for publication and will be formally accepted for publication once it meets all outstanding technical requirements.

Kind regards,

Yaodong Gu

Academic Editor

PLOS ONE

Additional Editor Comments (optional):

Well done!

Reviewers' comments:

Reviewer's Responses to Questions

**Comments to the Author**

1. If the authors have adequately addressed your comments raised in a previous round of review and you feel that this manuscript is now acceptable for publication, you may indicate that here to bypass the “Comments to the Author” section, enter your conflict of interest statement in the “Confidential to Editor” section, and submit your "Accept" recommendation.

Reviewer #1: All comments have been addressed

Reviewer #2: (No Response)

2. Is the manuscript technically sound, and do the data support the conclusions?

Reviewer #1: Yes

Reviewer #2: (No Response)

3. Has the statistical analysis been performed appropriately and rigorously? 

Reviewer #1: Yes

Reviewer #2: (No Response)

4. Have the authors made all data underlying the findings in their manuscript fully available?

Reviewer #1: Yes

Reviewer #2: (No Response)

5. Is the manuscript presented in an intelligible fashion and written in standard English?

Reviewer #1: Yes

Reviewer #2: (No Response)

6. Review Comments to the Author

Reviewer #1: The manuscript has been thoroughly reviewed, and all content-related concerns have been satisfactorily addressed. At this stage, the manuscript is well-prepared for publication, pending some final adjustments to meet formatting requirements. I recommend that the authors meticulously review the manuscript to ensure that all formatting aspects, including citations, figures, and tables, strictly adhere to the style guide. Attending to these final details will ensure the manuscript fully complies with publication standards.

Reviewer #2: Thanks to the efforts of the authors, the reviewer believes that this manuscript has reached the publication standard after revision.

7. PLOS authors have the option to publish the peer review history of their article (what does this mean?). If published, this will include your full peer review and any attached files.

Reviewer #1: No

Reviewer #2: **Yes: **Zixiang Gao

---

## [Editor Report · Acceptance letter]

22 Jul 2024

PONE-D-24-10897R1 

PLOS ONE

Dear Dr. Song, 

I'm pleased to inform you that your manuscript has been deemed suitable for publication in PLOS ONE. Congratulations! Your manuscript is now being handed over to our production team.

Kind regards, 

on behalf of

Professor Yaodong Gu 

Academic Editor

PLOS ONE